# Coinage-Metal Bond between [1.1.1]Propellane and M_2_/MCl/MCH_3_ (M = Cu, Ag, and Au): Cooperativity and Substituents

**DOI:** 10.3390/molecules24142601

**Published:** 2019-07-17

**Authors:** Ruijing Wang, Shubin Yang, Qingzhong Li

**Affiliations:** The Laboratory of Theoretical and Computational Chemistry, School of Chemistry and Chemical Engineering, Yantai University, Yantai 264005, China

**Keywords:** coinage-metal bonding, cooperativity, substituents, NBO, AIM

## Abstract

A coinage-metal bond has been predicted and characterized in the complexes of [1.1.1]propellane (P) and M_2_/MCl/MCH_3_ (M = Cu, Ag, and Au). The interaction energy varies between −16 and −47 kcal/mol, indicating that the bridgehead carbon atom of P has a good affinity for the coinage atom. The coinage-metal bond becomes stronger in the Ag < Cu < Au sequence. Relative to M_2_, both MCl and MCH_3_ engage in a stronger coinage-metal bond, both -Cl and -CH_3_ groups showing an electron-withdrawing property. The formation of coinage-metal bonding is mainly attributed to the donation orbital interactions from the occupied C-C orbital into the empty metal orbitals and a back-donation from the occupied *d* orbital of metal into the empty C-C anti-bonding orbital. In most complexes, the coinage-metal bond is dominated by electrostatic interaction, with moderate contribution of polarization. When P binds simultaneously with two coinage donors, negative cooperativity is found. Moreover, this cooperativity is prominent for the stronger coinage-metal bond.

## 1. Introduction

Small-ring organic molecules have fascinated organic chemists since they exhibit not only unique bonding properties but feature interesting reaction modes and stereochemistry as well [1]. [1.1.1]Propellane is the smallest member of this family but its electronic structure is still under debate [2,3] since its synthesis was reported in 1982 [4]. Jackson and Allen [2] performed an ab initio calculation on the nature of C_1_-C_3_ bond in [1.1.1]propellane (Scheme 1) and found this bond retains some σ-bridged π pattern. Based on the valence bond theory, Shaik and coworkers ascribed the bridge bond of [1.1.1]propellane to a charge shift (CS) bond [3]. Introducing such small-ring scaffolds can improve passive permeability, aqueous solubility, and metabolic stability, thus [1.1.1]propellane derivatives are extensively synthesized [5]. Very weak C=O⋅⋅⋅H–C hydrogen bonding involving propellane can be detected by atomic force microscopy [6]. Molecular electrostatic potential (MEP) map showed that the inverted-tetrahedral bridgehead atoms of the bridge bond of [1.1.1]propellane possess excess electron density, thus this molecule can be taken as a Lewis base to form a halogen bond with some halogen donor molecules [7].

Regium bond is a strong interaction that occurs between a σ-hole on a coinage-metal (regium) atom and an electron donor [8]. Before this name was proposed, the Lewis acidic character of a regium atom had long been recognized [9,10,11], where the corresponding interaction was called gold-bonding. Regium bonds between M_n_ clusters (M = Cu, Ag, Au; n = 2–6) and nucleophiles NH_3_ and HCN are in a large part dominated by Coulombic attraction, with a smaller orbital interaction contribution [12]. Other than molecules with lone-pair electrons, π molecules are also utilized to bind with regium nanoparticles [13,14] and gold compounds [15]. If MX (X = halogen) binds with benzene, a cation-π interaction is characterized since the oxidation state of M is +1 in MX [14]. Recently, Legon and Walker named this bond as a coinage-metal bond by analogy with the term ‘halogen bond’ [16]. Wang et al. compared the hydrogen, halogen, and coinage-metal bonds between small π molecules and HX/YX/MX (X = F, Cl, Br, I; Y = Cl, Br; M = Cu, Ag, Au) [17]. The distribution of σ-holes on the surface of nanostructured gold can be unveiled by MEPs [18]. The presence of σ-holes in nanoparticles of gold is helpful to explain its catalytic properties.

Cooperativity is an important property of noncovalent interactions since it largely determines the applications of noncovalent interactions in crystal engineering, molecular recognition, and biological functions [19,20,21]. Generally, both interactions are strengthened if the middle molecule acts as the Lewis acid and base, respectively. Both interactions are weakened when the middle molecule serves as a double Lewis acid/base. Coinage-metal bond also exhibits cooperativity with other types of interactions [11,22,23,24,25,26,27,28]. In FCCF⋅⋅⋅AgCCX⋅⋅⋅NCH (X = Cl, Br, I), both coinage-metal bond and halogen bond are simultaneously strengthened, although AgCCX is a double Lewis acid [23]. That is, coinage-metal bond sometimes displays some abnormal cooperativity.

In this paper, we studied the coinage-metal-bonded complexes between [1.1.1]propellane (P) and M_2_/MCl/MCH_3_ (M = Cu, Ag, Au). The coinage-metal bonds formed by M_2_ and MCl/MCH_3_ were compared to study the influence of substituents. The dependence of coinage-metal bonding strength on the nature of a coinage-metal atom was explored. The nature of the coinage-metal bond was unveiled by means of atoms in molecules (AIM), natural bond orbital (NBO), and energy decomposition (ED) analyses. Based on the binary Ag systems, six ternary systems of Ag_2_-P-Ag_2_, AgCH_3_-P-AgCH_3_, AgCl-P-AgCl, Ag_2_-P-AgCH_3_, Ag_2_-P-AgCl, and AgCH_3_-P-AgCl were designed to investigate the cooperativity of the coinage-metal bond.

## 2. Theoretical Methods

The geometries of the binary systems were first optimized at the MP2/aug-cc-pVDZ level. For the coinage atom, an aug-cc-pVDZ-PP [29] basis set was adopted to account for relativistic effects. To ensure that all structures corresponded to the true minima on the potential energy surfaces, harmonic frequency calculations were performed at the same level. The geometry optimization of the binary systems was also performed at the MP2/aug-cc-pVTZ(PP) and wB97X-D/aug-cc-pVTZ(PP) levels, where the integration grid of ωB97X-D method was 75,302 in Gaussian 09. The geometries of the ternary systems were only optimized at the wB97X-D/aug-cc-pVTZ(PP) level. Because MP2 neglects three-body intermolecular dispersion [30], the wB97X-D [31] method was used to study the structures and properties of all complexes. The interaction energies (∆E) with different methods were computed as the difference between the energy of the complex and the corresponding monomers with their geometries in that of the complex. A single-point energy calculation was also performed at the coupled-cluster singles and doubles augmented by a perturbative treatment of triple excitations (CCSD(T))/aug-cc-pVTZ(PP) level with the MP2/aug-cc-pVTZ(PP) geometries. Interaction energies were corrected for the basis set superposition error (BSSE) with the counterpoise method proposed by Boys and Bernardi [32]. All calculations were carried out with the Gaussian 09 program [33].

Molecular electrostatic potentials (MEPs) at the 0.001 electrons Bohr^−3^ isodensity surfaces were calculated via the WFA-SAS program [34] at the wB97X-D/aug-cc-pVTZ(PP) level. Natural bond orbital (NBO) analysis [35] was carried out at the HF/aug-cc-pVDZ(PP)//wB97X-D/aug-cc-pVTZ(PP) level to estimate orbital interactions and charge transfer (CT). The topological parameters of electron density, its Laplacian, and total energy density at the bond critical point (BCP) were carried out at the wB97X-D/aug-cc-pVTZ(PP) level with the AIM2000 [36] program. The interaction energy was decomposed into its five components by the GAMESS program [37] with the localized molecular orbital-energy decomposition analysis (LMOEDA) method [38] at the MP2/aug-cc-pVTZ(PP)//wB97X-D/aug-cc-pVTZ(PP) level.

## 3. Results and Discussion

### 3.1. Binary Systems

Figure 1 shows the optimized structures of P-M_2_, P-MCl, and P-MCH_3_ binary complexes at the wB97X-D/aug-cc-pVTZ(PP) level. All the geometries have C_3v_ symmetry with no imaginary frequencies. The M⋅⋅⋅C distance was much shorter than the sum of van der Waals (vdW) radii of the relevant atoms, indicating a strong attractive interaction between the two moieties. Though the vdW radius increased from Cu to Ag to Au, the M⋅⋅⋅C distance became longer from Cu⋅⋅⋅C to Au⋅⋅⋅C to Ag⋅⋅⋅C for each series of the complex. This inconsistency was primarily ascribed to the remarkably high first ionization energy and the electron affinity of Au. The M⋅⋅⋅C distance in P-MCl was shorter than that in the P-M_2_ analogue, which can be explained with the positive MEP on both M_2_ and MCl. The longer M⋅⋅⋅C distance in P-MCH_3_ than that in P-MCl was due to the weaker electron-withdrawing ability of the methyl group compared to the Cl atom.

The attractive interaction between both molecules can be understood with the MEP maps as shown in Figure 2. Obviously, there is a red region (σ-hole) along the M-M, M-Cl, and M-C bonds. The σ-hole was enlarged in sequence of Ag_2_ < Cu_2_ < Au_2_, AgCl < AuCl <CuCl, but AuCH_3_ < AgCH_3_ < CuCH_3_. The MEP of the σ-hole increased in the order M_2_ < MCH_3_ < MCl owing to the different electron-withdrawing ability among the Cl, CH_3_, and M. The bridgehead carbon atoms in P have negative MEPs (see the blue area of P in Figure 2). As a result, a coinage-metal bond formed between the bridgehead carbon atom of P and the M atom in M_2_/MCl/MCH_3_.

The interaction energy was computed with four methods including MP2/aug-cc-pVDZ(PP), MP2/aug-cc-pVTZ(PP), wB97X-D/aug-cc-pVTZ(PP), and CCSD(T)/aug-cc-pVTZ(PP), and the respective interaction energies were denoted as ∆E_MP2-pVDZ_, ∆E_MP2-pVTZ_, ∆E_wB97X_-_D-pVTZ_, and ∆E_CCSD(T)-pVTZ_. As shown in Table 1, the average deviation from the CCSD(T) results varied from 2.37 to 5.90 kcal/mol for the different systems. As expected, the larger basis set aug-cc-pVTZ(PP) brought out the larger interaction energy than the smaller basis set aug-cc-pVDZ(PP), and the difference was in a range of 1.8 to 5.7 kcal/mol, dependent on the different systems. Compared with ∆E_CCSD(T)-pVTZ_, the MP2/aug-cc-pVTZ(PP) method overestimated the interaction energy and this overestimation was even larger than 10 kcal/mol in P-Au_2_ and P-AuCl. The interaction energy calculated by the wB97X-D/aug-cc-pVTZ(PP) method was close to ∆E_CCSD(T)-pVTZ_ with comparison to the MP2 results. Furthermore, this method was successfully used to study coinage-metal bonds with small π molecules [17]. Thus, the wB97X-D/aug-cc-pVTZ(PP) data were used for discussion.

In all cases, the interaction energy had a large range from −16.34 kcal/mol to −47.30 kcal/mol. The interaction energy was closely related to the nature of the coinage metal atom. The Au complexes had the largest interaction energy and the Ag complexes had the smallest interaction energy. The MEP on the M atom can be partly responsible for the change of interaction energy in P-M_2_ since both terms linearly correlate. The MCl complexes show the stronger coinage-metal bond than the M_2_ analogues with a similar reason. The methyl group in MCH_3_ weakens the coinage-metal bond relative to MCl. The interaction energy in P-M_2_ approaches that in NH_3_-M_2_ [12], indicating [1.1.1]propellane is a good electron donor in coinage-metal bonding. It should be noted that the the oxidation state of M was different in M_2_ and MCl/MCH_3_, with 0 and +1, respectively [14].

Figure 3 shows the AIM diagrams of P-Cu_2_. A bond critical point (BCP) was present between the Cu atom and the bridgehead C atom of P, confirming the existence of a coinage-metal bond. The topological parameters at the BCP, including electron density, Laplacian, and total energy density, were collected in Table 2. The electron density ranging from 0.07 to 0.121 a.u. is partly out of the range suggested for non-covalent interactions [39]. Although the M⋅⋅⋅C BCP is different due to the M atom, its electron density had a consistent change with the interaction energy for each series of the complex. For the same M⋅⋅⋅C BCP, the electron density in P-MCl was larger than that in the P-M_2_ analogue. The electron density at the Au⋅⋅⋅C BCP in P-AuCH_3_ was smaller than that in P-Au_2_, while an opposite result is found for the electron density at the Ag⋅⋅⋅C BCP. Even so, both variations were consistent with the interaction energy. Therefore, the electron density could be used to estimate the strength of the coinage-metal bond. The value of Laplacian was in a range of 0.18 to 0.28 a.u., which was also out of the range suggested for non-covalent interactions [39]. This further confirms the existence of a strong coinage-metal bond.

The sign of Laplacian and total energy density can give some useful information for the nature of a non-covalent interaction. For all complexes, Laplacian is positive and energy density is negative, indicating that coinage-metal bonding is a partially covalent interaction [40]. The energy density was more negative for the stronger coinage-metal bond.

According to the NBO analyses, there are three main orbital interactions upon the formation of coinage-metal bonding: σ_C-C_ → σ*_M–M_/σ_C-C_ → σ*_M-Cl_/σ_C-C_ → σ*_M-C_, σ_C-C_ → LP*_M_, and LP_M_ → σ*_C-C_. The former two orbital interactions are donation orbital interactions from the occupied C-C orbital into the empty metal orbitals, while the latter orbital interaction is a back-donation from the occupied *d* orbital of metal into the empty C-C anti-bonding orbital. These orbital interactions are estimated with second-order perturbation; see Table 3. Obviously, the donation orbital interactions are stronger than the back-donation interaction. For σ_C-C_ → σ*_M-Cl_/σ_C-C_ → σ*_M-C_, it is stronger with the increase of the coinage-metal mass, while σ_C-C_ → σ*_M–M_ has consistent change with the interaction energy. For σ_C-C_ → LP*_M_, its change was consistent with the interaction energy only in the P-MCl complex. The back-donation orbital interaction showed an irregular change in the P-MCH_3_ complex but had an enhancing tendency for the heavier metal in the P-M_2_ and P-MCl complexes. For P-M_2_, the σ_C-C_ → LP*_M_ orbital interaction was stronger than the σ_C-C_ → σ*_M–M_ one, while the former orbital interaction was weaker than σ_C-C_ → σ*_M–C_ in P-MCH_3_. However, the relative magnitude of both donor orbital interactions was different for three complexes of P-MCl. Both σ*_M–M_/σ*_M–Cl_/σ*_M–C_ and LP*_M_ are related with the *d* anti-bonding orbitals of M, but we did not give any explanation for the above variations since no rule was found.

Accompanied with these orbital interactions, charge transfer (CT) occurred from the P to the M_2_/MCl/MCH_3_. This quantity was calculated as the sum of the charge on all atoms of P. For each series of complex, the charge transfer was the smallest in the Ag complex but largest in the Au complex. This sequence was similar to that of the interaction energy for each series of P-M_2_, P-MCl, or P-MCH_3_ complex. For the same coinage-metal donor, CT increases from P-M_2_ to P-MCH_3_ to P-MCl, which was generally consistent with the interaction energy. Of course, there was one exception between them in P-Au_2_ and P-AuCH_3_. Namely, the former complex had larger interaction energy than the latter, while the smaller CT was found in the former. Even so, their difference was not large.

To unveil the origin of the coinage-metal bonding, the interaction energy was decomposed into five terms: Electrostatic energy (E^ele^), exchange energy (E^ex^), repulsion energy (E^rep^), polarization energy (E^pol^), and dispersion energy (E^disp^), and the related results are shown in Table 4. The largest attractive term was from E^ex^ and this term was often offset by E^rep^, thus we only focused on other attractive terms of E^ele^, E^pol^, and E^disp^. For each complex, E^disp^ was smaller than both E^ele^ and E^pol^, but its contribution could not be ignored since its ratio to the sum of the three attractive energies was 14–22%. For most complexes, E^ele^ was larger than E^pol^, and the reverse result was found in P-AuCl. This indicated that most coinage-metal bonded complexes were dominated by electrostatic interaction, with moderate contribution from polarization interaction. When the complex varied from Cu to Ag to Au, each term was smallest in the Ag complex and largest in the Au complex for most complexes. However, some exceptions were also found. For instance, E^disp^ was smallest in P-AgCH_3_ but largest in P-CuCH_3_.

### 3.2. Ternary Systems

Figure 4 shows the optimized structures of six ternary systems of Ag_2_-P-Ag_2_, AgCH_3_-P-AgCH_3_, AgCl-P-AgCl, Ag_2_-P-AgCH_3_, Ag_2_-P-AgCl, and AgCH_3_-P-AgCl, which were used to study the cooperativity of the coinage-metal bond. Only the Ag ternary systems are discussed since most Cu and Au counterparts have imaginary frequencies. The binding distance showed a similar change in the different series of ternary systems. Namely, the binding distance in the ternary system was longer than that in the binary analogue. If the coinage donor was the same in the ternary system, both interactions would have an equal elongation. Moreover, this elongation became larger from AgCl-P-AgCl to Ag_2_-P-Ag_2_ to AgCH_3_-P-AgCH_3_. When the coinage donor was different in the ternary system, both interactions had an unequal elongation. Furthermore, this elongation was larger for the weaker coinage-metal bond. For example, this elongation was 0.016 Å for the stronger P-AgCl interaction but 0.050 Å for the weaker P-AgCH_3_ interaction.

The total interaction energy of the ternary system is listed in Table 5. The stability of the ternary system is similar to that of the corresponding binary system. A ternary system composed of two binary systems with stronger coinage-metal bonds is expected to be more stable. For instance, the stability of the ternary system increases from Ag_2_-P-Ag_2_ to AgCH_3_-P-AgCH_3_ to AgCl-P-AgCl. Both interactions are weakened in the ternary system since their interaction energies decrease (positive **∆∆E**). The stronger the coinage-metal bond is, the more it is weakened in the ternary system. The coinage-metal bond in Ag_2_-P decreases from 4.69 kcal/mol in Ag_2_-P-Ag_2_ to 6.14 kcal/mol in Ag_2_-P-AgCH_3_ to 7.45 kcal/mol in Ag_2_-P-AgCl. The similar weakening result is also found in other ternary systems. These results indicate that both coinage-metal bonds exhibit negative cooperativity in these Ag ternary systems. There is a repulsion force between two Ag donor molecules since their interaction energy is positive. This repulsion interaction increases from Ag_2_ to AgCH_3_ to AgCl.

The cooperativity was also estimated with cooperative energy (E_coop_), which was calculated with the formulas of E_coop_ = ∆E_total_ − ∆E_left_ − ∆E_right_ − ∆E_far_, where ∆E_total_ was the total interaction energy of a ternary system, ∆E_left_ the interaction energy of the optimized left dimer, ∆E_right_ the interaction energy of the optimized right dimer, and ∆E_far_ the interaction energy of two Ag donor molecules in the trimer. This term was positive, consistent with the negative cooperativity between both coinage-metal bonds. Similarly, E_coop_ increased from Ag_2_-P-Ag_2_ to AgCH_3_-P-AgCH_3_ to AgCl-P-AgCl, depending on the strength of coinage-metal bonding.

The change of the coinage-metal bonding strength in the ternary system can also be estimated with the electron density at the intermolecular BCP since they are relevant. Compared with the binary system, the electron density at the intermolecular BCP decreases (see Table 6), indicating the coinage-metal bond is weakened in the ternary system.

Considering the contribution of electrostatic interaction in the coinage-metal bond, the MEP on the free bridgehead carbon atom in the P binary complexes was examined. It was calculated to be 0.0445au in P-Ag_2_, 0.0469 a.u. in P-AgCH_3_ and 0.0590 a.u. in P-AgCl, respectively. It is of note that the negative MEP in the P monomer became positive in the P binary complexes. As a result, the second coinage-metal bond was unfavorable to be formed in views of electrostatic interaction.

It was interesting to find that the free bridgehead carbon atom with a positive MEP in the P binary complex still bound attractively with another coinage donor. This result can be partly attributed to the presence of orbital interactions between both molecules. These orbital interactions result in charge transfer from P to the coinage donor. This value decreased in the ternary system compared with that in the binary system (Table 7), consistent with the weakening of the coinage-metal bond. For the ternary complexes with two same coinage-metal bonds, the decrease of charge transfer grows up with the strengthening of the coinage-metal bond.

## 4. Conclusions

Nine binary systems of P-M_2_, P-MCH_3_, and P-MCl (M = Cu, Ag, and Au), as well as six Ag ternary systems have been investigated. The following conclusions are summarized as: (1) A coinage-metal bond is found between the bridgehead carbon atom of P and a coinage donor, with a partially covalent character. (2) The coinage-metal bonded complexes have big interaction energy of −16.3~ −47.3 kcal/mol, showing P is a good electron donor in the coinage-metal bond like that in H-bonding. (3) The strength of the coinage-metal bond is closely related to the nature of the coinage atom and becomes stronger in the Ag < Cu < Au order. (4) Both the electron-withdrawing group –Cl and –CH_3_ strengthen the coinage-metal bond. (5) The formation of the coinage-metal bonding is accompanied with the donor orbital interactions from the occupied C-C orbital into the empty metal orbitals and a back-donation from the occupied *d* orbital of metal into the empty C-C anti-bonding orbital. (6) Although the coinage-metal bond has a partially covalent character, electrostatic interaction is still dominant in most coinage-metal bonded complexes, and polarization and dispersion contributions are also important. (7) When the two bridgehead carbon atoms of P bind simultaneously with two coinage donors, negative cooperativity is found with more effect for the stronger coinage-metal bond.

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
