# Peer review of "Coinage-Metal Bond between [1.1.1]Propellane and M_2_/MCl/MCH_3_ (M = Cu, Ag, and Au): Cooperativity and Substituents"

_molecules, 2019, doi:10.3390/molecules24142601_

Round 1

Reviewer 1 Report

This contribution details an investigation into the intermolecular interaction (sometimes known as the regium bond or coinage metal bond) between small coinage metal complexes and the simplest propellane. The authors use a variety of computational techniques to investigate the interaction, with the conclusion that the complexes are predominately electrostatically bound with some polarization/charge transfer contribution. I find several problems with the manuscript, not least that the method used for the bulk of the work, MP2/aug-cc-pVDZ, is simply too low-level for a contemporary computational chemistry investigation. The basis set is far too small and MP2 is not a good choice for this type of intermolecular interaction. It overestimates dispersion, does not describe transition metals well, and is unreliable for three-body interactions (see the introduction to DOI:10.1063/1.4927304 for reference to this last point). The manuscript also does not include key references (both for this type of intermolecular interaction and the theoretical methods used). In its current form the manuscript should not be published as all calculations need to be redone with more reliable methods.

Some further points the authors should consider:

1) A recent article reviewed a lot of work on this type of interaction, covering a lot of experimental and theoretical work that the authors of the current work do not mention. See DOI: 10.1039/c8cp03432j  I note this review covers some of the same electrostatic potential work as the current contribution.

2) The description of the basis sets used is not sufficient - the aug-cc-pVDZ basis doesn't exist for Ag or Au. There is aug-cc-pVDZ-PP, which includes an ECP. Does the Cu basis include an ECP in the calculations (both versions are available)? Regardless, the original papers describing the basis sets should be cited.

3) A more minor point. The text refers to Figure 1 on page 81 when I believe this should be Fig 2. The opposite problem appears on Line 95.

Author Response

Reviewer 1

Comment 1: This contribution details an investigation into the intermolecular interaction (sometimes known as the regium bond or coinage metal bond) between small coinage metal complexes and the simplest propellane. The authors use a variety of computational techniques to investigate the interaction, with the conclusion that the complexes are predominately electrostatically bound with some polarization/charge transfer contribution. I find several problems with the manuscript, not least that the method used for the bulk of the work, MP2/aug-cc-pVDZ, is simply too low-level for a contemporary computational chemistry investigation. The basis set is far too small and MP2 is not a good choice for this type of intermolecular interaction. It overestimates dispersion, does not describe transition metals well, and is unreliable for three-body interactions (see the introduction to DOI:10.1063/1.4927304 for reference to this last point). The manuscript also does not include key references (both for this type of intermolecular interaction and the theoretical methods used). In its current form the manuscript should not be published as all calculations need to be redone with more reliable methods.

Response: In the revised manuscript, all results were again calculated with wB97XD/aug-cc-pVZT, where a bigger basis set and a method properly considering dispersion were used. This reference was also cited [22]. The key references have also been cited as [16] and [30].

These data were computed with four methods including MP2/aug-cc-pVDZ(PP), MP2/aug-cc-pVTZ(PP), wB97XD/aug-cc-pVTZ(PP), and CCSD(T)/aug-cc-pVTZ(PP), and the respective interaction energies are denoted as EMP2-pVDZ, EMP2-pVTZ, EwB97XD-pVTZ, and ECCSD(T)-pVTZ. ECCSD(T)-pVTZ was obtained with a single-point energy calculation on the MP2/aug-cc-pVTZ(PP) geometries. As expected, the larger basis set aug-cc-pVTZ(PP) brings out the larger interaction energy than the smaller basis set aug-cc-pVDZ(PP), and their difference is in a range of 1.8-5.7 kcal/mol, dependent on the different systems. Relative to the ECCSD(T)-pVTZ value, the MP2/aug-cc-pVTZ(PP) method overestimates the interaction energy and this overestimation is even larger than 10 kcal/mol in some systems. For wB97XD/aug-cc-pVTZ(PP) method, its value is close to ECCSD(T)-pVTZ with comparison to the MP2 results. Furthermore, this method was successfully used to study coinage-metal bonds with small π molecules [17]. Thus we applied the wB97XD/aug-cc-pVTZ(PP) data to discuss the corresponding results.

It was demonstrated that second-order Møller-Plesset perturbation theory (MP2) neglects three-body intermolecular dispersion [22].

Comment 2: A recent article reviewed a lot of work on this type of interaction, covering a lot of experimental and theoretical work that the authors of the current work do not mention. See DOI: 10.1039/c8cp03432j I note this review covers some of the same electrostatic potential work as the current contribution.

Response: This article was cited as Ref. 16.

Recently, Legon and Walker named the bond between MX and an electron donor as coinage-metal bond by analogy with the term halogen bond [16].

Comment 3: The description of the basis sets used is not sufficient - the aug-cc-pVDZ basis doesn't exist for Ag or Au. There is aug-cc-pVDZ-PP, which includes an ECP. Does the Cu basis include an ECP in the calculations (both versions are available)? Regardless, the original papers describing the basis sets should be cited.

Response: Yes, it is aug-cc-pVDZ-PP basis set. The corresponding reference was cited.

For the coinage atom, aug-cc-pVDZ-PP [30] basis set was adopted to account for relativistic effects.

Comment 4: A more minor point. The text refers to Figure 1 on page 81 when I believe this should be Fig 2. The opposite problem appears on Line 95.

Response: This has been corrected. Thanks

Reviewer 2 Report

The authors used M2/MCl/MCH3 (M = Cu, Ag, and Au) with [1.1.1]propellane as prototype systems to study regium bonds. Having obtained stable interaction energies, the bond was attributed by the authors to the donation from C-C to the metal orbitals complemented by back-donation from the metal to empty C-C s* orbitals. The authors also state that both -Cl and -CH3 groups show electron-withdrawing properties. Then, in contrast, the authors also state that the regium bond is dominated by electrostatic interactions (i.e, not orbital interactions).

I have many concerns with this manuscript (besides the aforementioned apparent contradiction). I believe there is a fundamental drawback in the author’s design of the systems and the subsequent interpretation. They are comparing M2 (oxidation state = 0) with MX (oxidation state = 1) and to both scenarios, the name “regium bond” is applied!  This is quite questionable (see e.g. the distinction between regium and cation interactions, Frontera et al, 10.1002/chem.201800820, and 10.3390/inorganics6030064 refs 13 e 14). A clear distinction between the two scenarios must be taken into account not only in the analysis of the results but also in the introduction, where references pertaining to both cases are indiscriminately used.  

I, therefore, recommend a major revision of the manuscript.  

Other issues:

* page 1, line 28  “Calculation on the nature of C1-C3 bond in ...” . A numbering scheme should be provided.

* page 1, line 37 “Regium bond is a stronger interaction”. Stronger than? This is a comparative of an adjective.

* page 1, line 40-42 “Regium bonds between Mn clusters (M = Cu, Ag, Au; n = 2–6) and nucleophiles NH3 and HCN are in large part dominated by Coulombic attraction, with a smaller orbital interaction contribution [12]” - reference 12 is not about this subject.

* page 2, lines 81-82 “Figure 1 shows the optimized structures of P-M2, P-MCl,and P-MCH3 binary complexes”. Figure 1 does not show the optimized geometries (shows the MEPs for the individual molecules). This should be Figure 2.

* page 2, lines 88-90 “The M⋅⋅⋅C distance in P-MCH3 is longer than that in the P-MCl counterpart since the methyl group has a weaker electron-withdrawing ability than the Cl atom”. Is the metyl group an electron-withdrawing ligand?

* page 3, Figure 1, the units are missing.

* page 3, lines 94-95 “The attractive interaction between both molecules can be understood with MEP maps of two monomers, shown in Figure 2” - this should be Figure 1.

* page 4, lines 107-108 “The MEP on the M atom can be partly responsible for the change of interaction energy”. Does the interaction energy correlate with the sigma-hole for each family?

* page 5, line 133 “There are three main orbital interactions upon the formation of regium bonding...” What type of analysis was performed to obtain these interactions? It is inferred from Table 2 that second-order perturbation energies from the NBO context are used, but this is only mentioned later (line 137).  

* When analyzing Table 2, it appears that for M-M, the largest contribution is from C-C --> LP* M; for M-Cl is similar apart from P-AuCl; while for M-CH3, the main contribution changes to C-C --> sigma* Cu–C. This is not clearly explained in the text nor a convincing interpretation of these changes when going from M-M (regium bonds) to the other interaction (M-Cl vs M-CH3). How were the charge transfer values calculated?

* page 6, lines 148-150 “These inconsistencies are partly explained with the coexistence of orbital interactions with a reverse direction of charge shift” - what evidences are presented for this claim?

* the English must also be improved in a subsequent submission.

Author Response

Reviewer 2

The authors used M2/MCl/MCH3 (M = Cu, Ag, and Au) with [1.1.1]propellane as prototype systems to study regium bonds. Having obtained stable interaction energies, the bond was attributed by the authors to the donation from C-C to the metal orbitals complemented by back-donation from the metal to empty C-C s* orbitals. The authors also state that both -Cl and -CH3 groups show electron-withdrawing properties. Then, in contrast, the authors also state that the regium bond is dominated by electrostatic interactions (i.e, not orbital interactions).

Comment 1: I have many concerns with this manuscript (besides the aforementioned apparent contradiction). I believe there is a fundamental drawback in the author’s design of the systems and the subsequent interpretation. They are comparing M2 (oxidation state = 0) with MX (oxidation state = 1) and to both scenarios, the name “regium bond” is applied! This is quite questionable (see e.g. the distinction between regium and cation interactions, Frontera et al, 10.1002/chem.201800820, and 10.3390/inorganics6030064 refs 13 e 14). A clear distinction between the two scenarios must be taken into account not only in the analysis of the results but also in the introduction, where references pertaining to both cases are indiscriminately used. I, therefore, recommend a major revision of the manuscript.

Response: It is right. This has been updated. This bond is written as coinage-metal bond according to suggestion in Ref [6]. The following sentences have been added.

If MX (X= halogen) binds with benzene, a cation-π interaction is characterized since oxidation state of M is +1 in MX [14].

It should be noted that oxidation state of M is different in M2 and MCl/MCH3, with 0 and +1, respectively [14].

Comment 2: page 1, line 28 “Calculation on the nature of C1-C3 bond in ...” . A numbering scheme should be provided.

Response: This scheme with number has been added.

Comment 3: page 1, line 37 “Regium bond is a stronger interaction”. Stronger than? This is a comparative of an adjective.

Response: This has been corrected as: Regium bond is a strong interaction.

Comment 4: page 1, line 40-42 “Regium bonds between Mn clusters (M = Cu, Ag, Au; n = 2–6) and nucleophiles NH3 and HCN are in large part dominated by Coulombic attraction, with a smaller orbital interaction contribution [12]” - reference 12 is not about this subject.

Response: Ref. 12 has been updated.

Comment 5: page 2, lines 81-82 “Figure 1 shows the optimized structures of P-M2, P-MCl, and P-MCH3 binary complexes”. Figure 1 does not show the optimized geometries (shows the MEPs for the individual molecules). This should be Figure 2.

Response: This error has been corrected.

Comment 6: page 2, lines 88-90 “The M⋅⋅⋅C distance in P-MCH3 is longer than that in the P-MCl counterpart since the methyl group has a weaker electron-withdrawing ability than the Cl atom”. Is the methyl group an electron-withdrawing ligand?

Response: It is known that carbon has the greater electronegativity than the coinage metal, thus we think the methyl group is an electron-withdrawing group here. The negative charge on the Cl atom is larger than that on the methyl group, thus the methyl group is a weaker electron-withdrawing ability than the Cl atom.

Comment 7: page 3, Figure 1, the units are missing.

Response: The units have been marked.

Comment 8: page 3, lines 94-95 “The attractive interaction between both molecules can be understood with MEP maps of two monomers, shown in Figure 2” - this should be Figure 1.

Response: This error has been corrected.

Comment 9: page 4, lines 107-108 “The MEP on the M atom can be partly responsible for the change of interaction energy”. Does the interaction energy correlate with the sigma-hole for each family?

Response: According to the data with the new method, this sentence has been modified as: The MEP on the M atom can be partly responsible for the change of interaction energy in P-M2 since both terms linearly correlate.

Comment 10: page 5, line 133 “There are three main orbital interactions upon the formation of regium bonding...” What type of analysis was performed to obtain these interactions? It is inferred from Table 2 that second-order perturbation energies from the NBO context are used, but this is only mentioned later (line 137).

Response: The sentence of “According to the NBO analyses” has been added.

Comment 11: When analyzing Table 2, it appears that for M-M, the largest contribution is from C-C --> LP* M; for M-Cl is similar apart from P-AuCl; while for M-CH3, the main contribution changes to C-C --> sigma* Cu–C. This is not clearly explained in the text nor a convincing interpretation of these changes when going from M-M (regium bonds) to the other interaction (M-Cl vs M-CH3). How were the charge transfer values calculated?

Response: The explanation of calculating charge transfer was given.

This quantity was calculated as the sum of charge on all atoms of P.

For P-M2, the σC-C→LP*M orbital interaction is stronger than the σC-C→σ*M–M one, while the former orbital interaction is weaker than σC-C→σ*M–C in P-MCH3. However, the relative magnitude of both donor orbital interactions is different for three complexes of P-MCl. We think both σ*M–M/σ*M–Cl/σ*M–C and LP*M are related with the d anti-bonding orbitals of M. However, the type of the d orbital is difficult to be differentiated in the NBO analyses and thus we did not give a rational explanation for the above variations.

Comment 12: page 6, lines 148-150 “These inconsistencies are partly explained with the coexistence of orbital interactions with a reverse direction of charge shift” - what evidences are presented for this claim?

Response: Such inconsistencies are not found with the new method, thus this sentence has been replaced by “This sequence is similar to that of interaction energy for each series of P-M2, P-MCl or P-MCH3 complex. For the same coinage metal donor, CT increases from P-M2 to P-MCH3 to P-MCl, which is generally consistent with the interaction energy. Of course, there is one exception between them in P-Au2 and P-AuCH3. Namely, the former complex has the larger interaction energy than the latter, while smaller CT is found in the former. Even so, their difference is not large”.

Comment 13: the English must also be improved in a subsequent submission.

Response: English has been checked. 

Round 2

Reviewer 1 Report

In this revised version the authors have addressed all of the suggestions of the referees, and a much stronger manuscript is the result. In particular, the switch to using a modern DFT functional and testing against CCSD(T) level results makes the outcomes much more convincing.

I have two further suggestions before the manuscript is accepted for publication.

1) The authors use a functional they term wB97XD. I believe this should be ωB97X-D and the reference for this functional must be included.

2) The integration grid used in the DFT calculations should be specified.

Author Response

Response for the comments of the reviewer 1

In this revised version the authors have addressed all of the suggestions of the referees, and a much stronger manuscript is the result. In particular, the switch to using a modern DFT functional and testing against CCSD(T) level results makes the outcomes much more convincing. I have two further suggestions before the manuscript is accepted for publication.

Comment 1: The authors use a functional they term wB97XD. I believe this should be ωB97X-D and the reference for this functional must be included.

Response: It is right. We have corrected this error in the whole text. The Ref. 31 has been cited.

Comment 2: The integration grid used in the DFT calculations should be specified.

Response: We used ωB97X-D method in Gaussian 09 to perform all calculations and all parameters are in default, thus the integration grid is 75302. The following sentence has been added in the method.

where the integration grid of ωB97X-D method is 75302 in Gaussian 09. 

Reviewer 2 Report

The revised version of the manuscript tackled most of the issues raised in my previous report. However, some details still prevail (see bellow):

* Lines 56, 57 “It was demonstrated that second-order Møller-Plesset perturbation theory (MP2) neglects three-body intermolecular dispersion [22]”. This sentence is a justification concerning the choice of the method. It belongs to the “Theoretical Methods” section. Something like, since “since MP2 neglects three-body intermolecular dispersion, we used ...”

* The changes introduced in the methods section are quite confusing. So, the structures were first optimized at the MP2/aug-cc-pVDZ level and eventually lead to ∆EMP2-pVDZ values. Then, it is stated that these binary systems were then optimized again on the MP2/aug-cc-pVDZ(PP) geometries at the MP2/aug-cc-pVTZ(PP) and wB97XD/aug-cc-pVTZ(PP) levels (yielding ∆E MP2-pVTZ and ∆E wB97XD-pVTZ). The sentence is misleading. Were the geometries reoptimized? If this is so, it suffices to say that “these binary systems were also optimized at the MP2/aug-cc-pVTZ(PP) and wB97XD/aug-cc-pVTZ(PP) levels”. In this section, a mention to the CCSD(T) calculations (how were they performed) should also be provided (instead of the explanation in the main text).

* Lines 91,92 - “at the MP2/aug-cc-pVTZ(PP)/wB97XD/aug-cc-pVTZ(PP) level “. The double slash // is missing.  

* Line 94 “Figure 1 shows the optimized structures”. At which level of theory? The same is applied to Figure 1 Caption. Apparently, the structures were optimized using different levels of theory according to the theoretical methods. How does the geometry change among methods?

* Table 1 - A line containing the average deviation from CCSD(T) results would be helpful.  

* Lines 178-179 - “However, the type of the d orbital is difficult to be differentiated in the NBO analyses and thus we did not give a rational explanation for the above variations”. This is not true because it is possible to obtain a listing of NBOs displaying the form and occupancy of the complete set of orbitals that span the input AO space.

* The English still needs improvement.  

Author Response

The revised version of the manuscript tackled most of the issues raised in my previous report. However, some details still prevail (see bellow):

Comment 1: Lines 56, 57 “It was demonstrated that second-order Møller-Plesset perturbation theory (MP2) neglects three-body intermolecular dispersion [22]”. This sentence is a justification concerning the choice of the method. It belongs to the “Theoretical Methods” section. Something like, since “since MP2 neglects three-body intermolecular dispersion, we used ...”

Response: This sentence has been moved into the Theoretical method and modified as: Since MP2 neglects three-body intermolecular dispersion [30], we thus used wB97X-D [31] method to study the structures and properties of all complexes.

Comment 2: The changes introduced in the methods section are quite confusing. So, the structures were first optimized at the MP2/aug-cc-pVDZ level and eventually lead to ∆EMP2-pVDZ values. Then, it is stated that these binary systems were then optimized again on the MP2/aug-cc-pVDZ(PP) geometries at the MP2/aug-cc-pVTZ(PP) and wB97XD/aug-cc-pVTZ(PP) levels (yielding ∆E MP2-pVTZ and ∆E wB97XD-pVTZ). The sentence is misleading. Were the geometries reoptimized? If this is so, it suffices to say that “these binary systems were also optimized at the MP2/aug-cc-pVTZ(PP) and wB97XD/aug-cc-pVTZ(PP) levels”. In this section, a mention to the CCSD(T) calculations (how were they performed) should also be provided (instead of the explanation in the main text).

Response: All structures of binary systems were optimized with three methods including MP2/aug-cc-pVDZ(PP), MP2/aug-cc-pVTZ(PP) and wB97X-D/aug-cc-pVTZ(PP). These sentences have been modifies as: The geometry optimization of binary systems was also optimized at the MP2/aug-cc-pVTZ(PP) and wB97X-D/aug-cc-pVTZ(PP) levels.

A mention to the CCSD(T) calculations was given in the method: A single-point energy calculation was also performed at the CCSD(T)/aug-cc-pVTZ(PP) level with the MP2/aug-cc-pVTZ(PP) geometries.

Comment 3: Lines 91,92 - “at the MP2/aug-cc-pVTZ(PP)/wB97XD/ aug-cc-pVTZ(PP) level “. The double slash // is missing.

Response: This has been corrected as at the MP2/aug-cc-pVTZ(PP)//wB97XD/ aug-cc-pVTZ(PP) level

Comment 4: Line 94 “Figure 1 shows the optimized structures”. At which level of theory? The same is applied to Figure 1 Caption. Apparently, the structures were optimized using different levels of theory according to the theoretical methods. How does the geometry change among methods?

Response: This method has been added as: at the wB97X-D/aug-cc-pVTZ(PP) level. The C3,v symmetry of binary systems is not changed for three different methods. If the different methods are used, the binding distances are changed. However, their change is like for the different methods.

Comment 5:  Table 1 - A line containing the average deviation from CCSD(T) results would be helpful.

Response: This has been added in Table 1. The following sentence was added: We also listed the average deviation from the CCSD(T) results. This deviation varies from 2.37 to 5.90 kcal/mol.

Comment 6: Lines 178-179 - “However, the type of the d orbital is difficult to be differentiated in the NBO analyses and thus we did not give a rational explanation for the above variations”. This is not true because it is possible to obtain a listing of NBOs displaying the form and occupancy of the complete set of orbitals that span the input AO space.

Response: This sentence is written as: Both σ*M–M/σ*M–Cl/σ*M–C and LP*M are related with the d anti-bonding orbitals of M, but we did not give any explanation for the above variations since no rule is found.

Comment 7: The English still needs improvement.

Response: We have checked it.